# Sb₂Se₃ Polycrystalline Thin Films Grown on Different Window Layers

**Stefano Pasini** [1,*], **Donato Spoltore** [1], **Antonella Parisini** [1], **Gianluca Foti** [1], **Stefano Marchionna** [2], **Salvatore Vantaggio** [1], **Roberto Fornari** [1] **and Alessio Bosio** [1]

1  Department of Mathematical, Physical and Computer Science, University of Parma, 43124 Parma, Italy
2  Ricerca Sistema Energetico-RSE Spa, 20134 Milano, Italy
*  Correspondence: stefano.pasini@unipr.it

**Abstract:** Sb₂Se₃ is a typical V₂VI₃ binary chalcogenide compound characterized by a single crystalline phase and a fixed composition. Sb₂Se₃ displays a narrow energy gap ranging from 1.1 to 1.3 eV, which are quite optimal values for single-junction solar cells. Earth-abundant and non-toxic components make this material a good candidate for low-cost thin-film solar cells. In substrate configuration, a world record efficiency of 9.2% was recently obtained. Sb₂Se₃ thin films exhibit an accentuated predisposition to form $(Sb_4Se_6)_n$ ribbons along the [001] direction. This anisotropy heavily influences the charge transport of the photogenerated carriers. In this work, structural characterization of the Sb₂Se₃ films showed that the crystalline quality and preferential orientation are strongly dependent on the window layer used. To better understand the growth mechanism, Sb₂Se₃ thin films were deposited by close-spaced sublimation on five different window layers, such as CdS, CdS:F, CdSe, As₂S₃, and ZnCdS. Sb₂Se₃-based solar cells, realized in superstrate configuration on these different substrates, evidently demonstrate the influence of the Sb₂Se₃ preferential orientation on the photovoltaic parameters.

**Keywords:** solar cells; thin film; Sb₂Se₃; CdS; ZnCdS; texture coefficient





## 1. Introduction

Antimony selenide (Sb₂Se₃) is a very promising material for developing innovative high-efficiency thin film solar cells, since it is based on abundant and less toxic elements on the Earth's crust than other materials suitable for the same application, such as CdTe and CIGS [1]. Moreover, it presents a good absorption coefficient ($\alpha > 10^5$ cm$^{-1}$) for the visible part of the solar spectrum [2], acceptable room temperature (RT) carrier mobility ($\mu_e \approx 15$ cm²V$^{-1}$ s$^{-1}$ and $\mu_h \approx 5.1$ cm²V$^{-1}$ s$^{-1}$) [3–5], and a suitable band gap ($E_g \approx 1.17$ eV) [6] that falls near the maximum of the Schottky-Quiesser limit [7]. This material crystallizes in the orthorhombic structure, and it presents a unique quasi-one-dimensional structure, formed by $((Sb_4 - Se_6)_n)$ chains (ribbons) along the c-axis that are bounded by weak van der Walls forces [8].

The identification of a good window layer as an n-type partner coupled with a p-type Sb₂Se₃, plays a crucial role to obtain high efficiency solar cells. In the last few years, different candidates have been checked to fill the gap; for example, Zinc oxide (ZnO) [9] and Titanium dioxide (TiO₂) [10] window layers were tested in Sb₂Se₃-based solar devices, reaching 5.93% and 5.5% power conversion efficiency (PCE), respectively. A world record PCE of 9.2% was achieved using cadmium sulphide (CdS) as the window layer in a substrate configuration solar cell, while a 7.6% record PCE was obtained in a superstrate configuration [11,12].

An optimal window layer should maximize the growth of the antimony selenide grains orthogonally to the substrate plane, such that the growth is oriented along the c-axis of the Sb₂Se₃ unit cell. This preferential spatial arrangement reduces the carrier

recombination at the interface between two ribbons [13,14] during the normal operation of the final photovoltaic device. This requirement is due to the peculiar structure of $Sb_2Se_3$: since $(Sb_4 - Se_6)_n$ ribbons interact with a strong covalent bond along [001], this guarantees a good carrier transport in that direction, while opposite behaviors are true for [100] or [010] directions, along which ribbons are bound by weak Van der Waals forces [13].

As far as we know from the literature, the Close-Spaced Sublimation (CSS) technique has already been widely used to deposit $Sb_2Se_3$ films in superstrate configuration thin-film solar cells. Despite this, a systematic study of how $Sb_2Se_3$ grows on different window layers has not yet been performed. For this reason, to study the effect of high-temperature deposition on the physical properties of $Sb_2Se_3$ films, the substrate has been varied as a CSS deposition parameter.

In this framework, five candidates used as window layers for $Sb_2Se_3$-based thin film solar cell in superstrate configuration (Figure 1a) were studied: cadmium sulphide (CdS), fluorine-cadmium sulphide (CdS:F), cadmium selenide (CdSe), zinc-cadmium sulphide ($Zn_{0.15}Cd_{0.85}S$) and arsenic sulphide ($As_2S_3$).

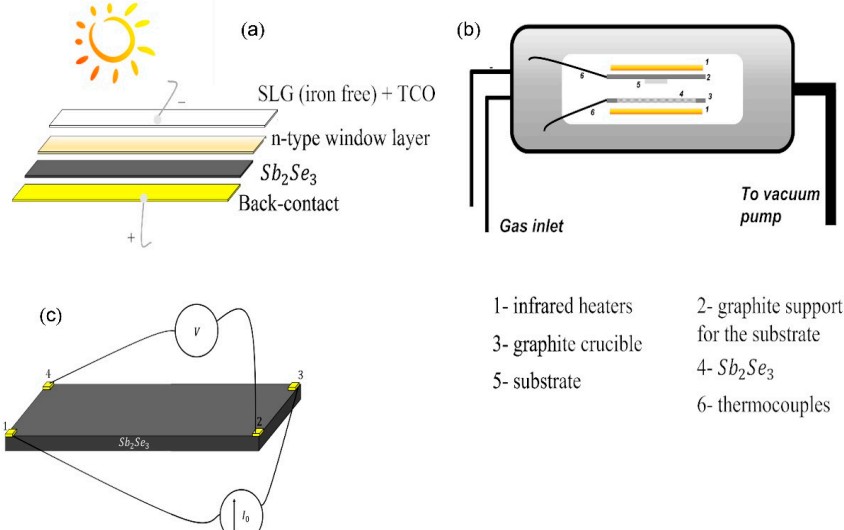

**Figure 1.** (**a**) The superstrate structure of the solar cells; (**b**) Sketch of CSS system; (**c**) van der Pauw measurement configuration.

Cadmium sulphide was considered because it has been successfully used for the same application in other thin-film solar cell technologies (CdTe and CIGS). Moreover, the actual world record PCE for $Sb_2Se_3$-based solar cells has been achieved using CdS as window layer. For this reason, solar cells realized with CdS represent a good reference for all the other devices based on the different investigated window layers [12,15,16].

One of the main problems of using the CdS as a window layer for $Sb_2Se_3$ remains the lattice mismatch with the absorber that could be minimized with the introduction of a double buffer layer [17].

In the literature, it has also been reported that the presence of $CdF_2$ into the grain boundaries of CdS promotes the chemical stability of the material [18]. For exploiting this advantage, cadmium sulphide in an atmosphere of Ar + $CHF_3$ as sputtering process gas was deposited.

Selenium vacancies are generated during the $Sb_2Se_3$ deposition at substrate temperature of 360 °C. These vacancies could act as deep donor levels, lowering the photovoltaic parameters [19,20]. To verify the possible filling of selenium vacancies, CdSe was tested to encourage the interaction with the growing $Sb_2Se_3$ film, which could be effective for the diffusion of Se atoms.

Arsenic sulphide was used to exploit the quasi-rheotaxy growth of the film [21,22]. In the quasi-rheotaxy approach, atoms constituting the surface layers of any material move

as if they were in a liquid state at temperatures up to 30% lower than the melting point. The material therefore does not melt, but the surface layers appear to be melted. This can give a high surface mobility to the adsorbed atoms and to the forming clusters. The high cluster mobility facilitates both coalescence and orientation, favoring large crystalline grain growth [23,24], and this is even more true in the case of a glassy system such as $As_2S_3$ [25]. Quasi-rheotaxy preserves the advantages of rheotaxy, which consists of growing thin films on liquid surfaces, but it avoids its main disadvantage, the formation of droplets.

By introducing an appropriate percentage of zinc inside the CdS lattice, the transparency of the window layer in the visible region is increased. At the same time, Zn changes both the band alignment with antimony selenide and the resistivity that is increased [26]. A 15 at% of Zn has been estimated to be an optimal value to maximize transparency and make the resistivity of the window layer suitable for photovoltaic applications.

Starting from the collected data on the effects of the different compounds tested, the final goal of this work is to describe the preferential growth orientation of the $Sb_2Se_3$ crystalline grains along the (002) plane (perpendicular to the ribbons axis) driven by the interaction with the window layers.

The obtained results show which are the best substrate for the $Sb_2Se_3$ growing, since there is a strict correlation between the photovoltaic parameters of the cells and the directions along which the ribbons grow. Probably, any improvement that the solar cell will exhibit in the future will be achieved by taking these results into account.

From this point of view, the strategy used is not limited to photosensitive devices but can be effective whenever one material is deposited on top of another. If the deposited material is strongly anisotropic, such as structures with reduced dimensions (1D and 2D), the method used in this work can give interesting results in the analysis of crystallinity and preferential orientations. This methodology can be used for the structural characterization of nanostructure-based devices such as nanoplateled LEDs [27], nanostructured solar cells [28], and more generally semiconductor nanodevices [29–34].

## 2. Materials & Methods

All the window layers were grown by low-temperature radiofrequency (r.f.) magnetron sputtering (MS) in a working Ar gas pressure of $5 \times 10^{-1}$ Pa except for sample B, for which a $CHF_3$ partial pressure of $5 \times 10^{-3}$ Pa was also introduced.

Samples with different window layers are indexed with capital letters, and the deposition parameters are described in Table 1.

**Table 1.** The main sputtering parameters related to the deposition of the window layers.

| Material (Working Atmosphere) | Substrate Temperature [°C] | Power Density [W/cm$^2$] | Deposition Rate [Å/s] | Thickness [Å] | Sample |
|---|---|---|---|---|---|
| - | 250 | 0.7 | 4 | 3000 | A |
| CdS (Ar + CHF$_3$) | 250 | 0.9 | 4 | 3000 | B |
| CdS + CdSe (Ar) | 250 | (0.7) (0.6) | (4) (4.4) | 3000 + 500 | C |
| CdS + As$_2$S$_3$ (Ar) | 220 | (0.7) (0.6) | (4) (2.4) | 3000 + 500 | D |
| CdS + ZnCdS (Ar) | 220 | (0.7) (0.8) | (4) (4) | 600 + 300 | E |

Sample A: CdS film, 300 nm thick, deposited by sputtering at a temperature of 200 °C is characterized by a direct energy gap of 2.42 eV with a hexagonal crystal structure, which becomes cubic if annealed over 400 °C [35].

Sample B: CdS:F film, 300 nm thick, is deposited by sputtering at 200 °C in an Ar + $CHF_3$ starting with a CdS target. The CdS:F film is characterized by a direct energy gap of 2.85 eV and a hexagonal crystal structure [36].

Sample C: CdSe film, 50 nm thick, deposited by sputtering at 200 °C in pure Ar, is characterized by a direct energy gap of 1.74 eV with a hexagonal crystal structure [37].

Sample D: As2S3 film, 50 nm thick, deposited by sputtering at 200 °C in pure Ar, is characterized by a direct energy gap of 2.35 eV with a monoclinic crystal structure [38].

Sample E: CdS + $Zn_{0.15}Cd_{0.85}S$ (ZnCdS) films, in which the CdS film is the same as in sample A and the ZnCdS film, 50 nm thick, is deposited by sputtering at 200 °C in pure Ar. ZnCdS film with a direct energy gap of 2.64 eV with a hexagonal crystal structure [39].

The thicknesses of all these films were optimized to avoid a significant blue light absorption (e.g., $E_{gCdS} \sim 2.42$ eV) and to prevent pinhole formation, which would affect the p-n junction behaviour [40–42].

Antimony selenide films were deposited by the close-spaced-sublimation (CSS) technique (Figure 1b) [43] using an argon partial pressure in the range 10–20 Pa. The crucible was heated at a temperature of $T_c = 550$ °C, and the substrate, even though not directly heated by an external source, reached a temperature of $T_s = 360$ °C due to convection and irradiation heating from the facing crucible. The typical distance between the crucible and substrate in CSS is $(2–4) \times 10^{-3}$ m. Since this distance is less than the mean free path of the sublimated particles, they do not experience scattering with the inert gas (Argon), and this condition grants fast, uniform thin film growth and high crystalline quality.

A substantial modification, which consists in the use of a compact block of $Sb_2Se_3$, previously melted and resolidified, instead of $Sb_2Se_3$ granules, was implemented. The use of a compact block of $Sb_2Se_3$ is effective in getting a uniform heating of the material, avoiding any burst of microparticles from the source to the substrate. By using this CSS system, in 5 min deposition an $Sb_2Se_3$ film with a thickness of $\sim 4\mu m$ is obtained.

XRD measurements were performed using a linear detector LYNXEYE (Bruker, Karlsruhe, Germany) and the crystalline phase identification was carried out by the PDF4 + database (ICDD, Newtown Square, PA, USA).

Raman analyses were performed using a He-Ne laser with line emission at 632.8 nm. Laser is focused on samples, (similar to a finished cell in which only the layer that constitutes the back contact is missing) in a nearly backscattered geometry with a HORIBA-Jobin Yvon LabRam confocal micro-spectrometer (300 mm focal length spectrograph) equipped with an integrated Olympus BX40 microscope, with $4\times$, $10\times$, $50\times$ Ultra Long Working Distance (ULWD) and $100\times$ objectives. In order to perform a correct measurement, the set-up was calibrated using the 520.6 $cm^{-1}$ Raman peak of silicon. Experimental data were manipulated by using LabSpec5 built-in software. The $Sb_2Se_3$ film are illuminated in confocal configuration, which means that only this layer is interested in the Raman measurement.

From the electrical point of view, on $Sb_2Se_3$ films, a four-point resistivity measurement was performed. In this case, $Sb_2Se_3$ was grown on an insulating material, such as sputtered zinc oxide, and the electrical contacts, formed by a 200 nm-thick $Sb_2Te_3$ film covered by a 100 nm-thick Pt film, were fabricated by a radiofrequency magnetron sputtering in the van der Pauw configuration [44] on the four corners of square-shaped samples having area $\sim 1$ $mm^2$ (Figure 1c). The measurement was performed by injecting and extracting current into a pair of contacts using a Keithley 220 programmable current source instrument, while the electrical potential between the other pair of contacts was read by means of a Keithley 617 programmable electrometer.

The complete $Sb_2Se_3$-based solar cell with 1 $cm^2$ of active area, realized in superstrate configuration, starting from a low-iron soda lime glass as a substrate, is made by the following layers:

1.  n-type part (all these films are MS deposited)

    a. Transparent Conducting Oxide (TCO) made up of an 800 nm thick ITO film
    b. high resistivity transparent (HRT) ZnO layer, 150 nm thick
    c. CdS, CdS:F, CdS + CdSe, CdS + $As_2S_3$, CdS + ZnCdS films representing the different window layers used for testing, whose thicknesses are described in Table 1.

2.  p-type part

    a. Absorber layer–$Sb_2Se_3$ film, 4–5 µm thick (CSS-deposited)

b.   Back contact–Sb$_2$Te$_3$ film, 200 nm thick covered by Pt film, 100 nm thick (both films are MS deposited)

The I-V characteristics of complete-solar cells were obtained by using a continuous LOT-Oriel solar simulator (Oriel, Irvine, CA, USA), equipped with an air mass AM1.5 filter and with a 1 KW/m$^2$ light power density supplied by a 600 W Xenon lamp (Oriel, Irvine, CA, USA). A calibrated pyranometer was used as a reference, and the measurements were made at the standard temperature of 298 K. Short-circuit current density ($J_{sc}$) values, non-corrected for the spectral mismatch, were measured over a calibrated shunt resistor supplied by a Keithley 4200-SCS instrument (Tektronix, Solon, OH, USA).

## 3. Results and Discussion

### 3.1. Structural Characterization

The reference card used to analyse the experimental XRD pattern of Sb$_2$Se$_3$ thin film is JCPDS 15-0861, which refers to Pbnm Sb$_2$Se$_3$ space group, as shown in Figure 2 by using the VESTA tool [45].

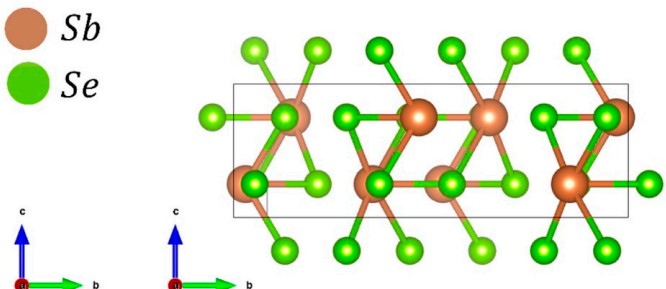

**Figure 2.** Spatial representation of atoms in Sb$_2$Se$_3$ crystal structure referred to orthorhombic Pbnm space group.

To evaluate the preferential orientation of the Sb$_2$Se$_3$ crystalline grains, the most representative planes in the XRD patterns reported in Figure 3 ((221), (301), (211), (002), (310), (212), (041), and (141)) were selected to estimate the texture coefficient ($TC_{hkl}$) associated with the (hkl) plane (following [46]])

$$TC_{hkl} = \frac{I_{hkl}/I_{0hkl}}{\sum_{i=0}^{n} I_{h_ik_il_i}/I_{0h_ik_il_i}} \cdot 100\%$$

where $I_{hkl}$ is the experimental peak intensity related to the selected plane, $I_{0hkl}$ is the intensity of the same plane reported in the reference card and n is the total number of chosen planes.

By comparing these XRD patterns with the reference card JCPDS 15-0861, it is evident that in the analysed Sb$_2$Se$_3$ films there aren't secondary phases, in agreement with the Raman spectra reported in Figure 4. RT Raman analysis shows that three peaks exist in all samples at 155 , 192  and 212 cm$^{-1}$ that corresponds to B$_{xg}$, and A$_g$ Raman active vibrational modes [47].

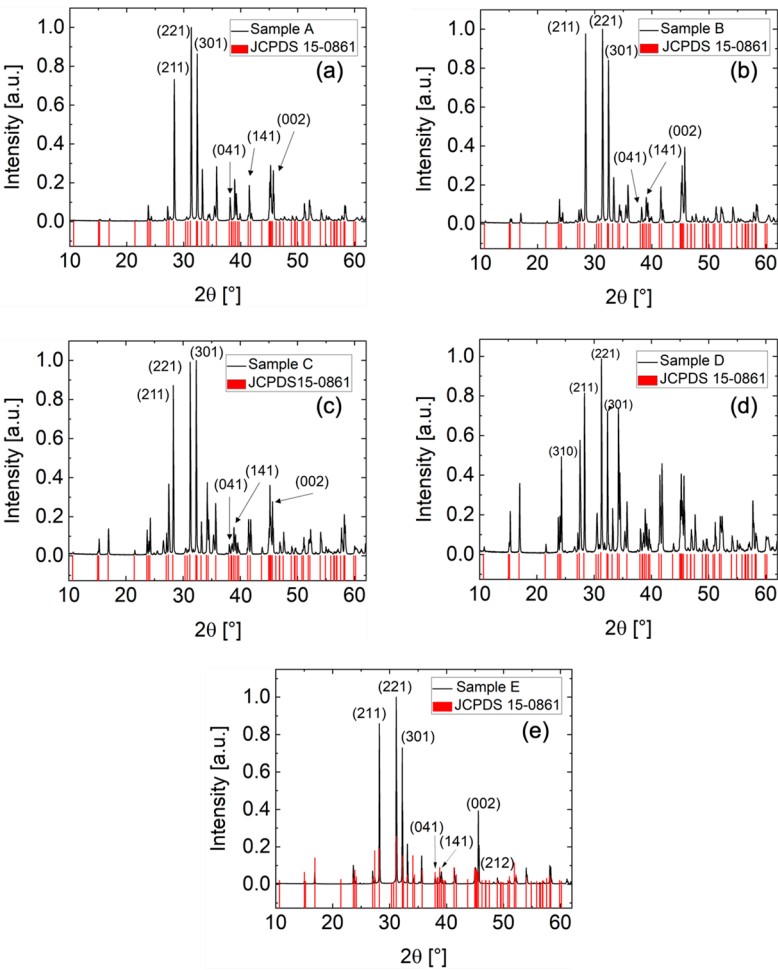

**Figure 3.** XRD spectra of Antimony selenide films grown on (**a**) CdS, (**b**) CdS:F, (**c**) CdSe, (**d**) As$_2$S$_3$ and (**e**) ZnCdS.

By imposing a threshold of 10% as the minimum TC for every plane taken into consideration from the XRD [48], it is evidenced in Figure 5a that the investigated samples show diffraction peaks corresponding to the (301), (221), (002), and (211) planes.

In particular, the analysis of texture coefficient values shows that there are preferential reflections from different planes, namely (301) for samples A (CdS) and C (CdSe), while for samples B (CdS:F) and E (ZnCdS) it is (002).

Since this material crystallizes in the orthorhombic phase, with $(Sb_4Se_6)_n$ ribbons stacked in parallel in the [001] direction [6], this is believed to be the direction in which charge transport is favoured [47,49]. Sample D (As$_2$S$_3$), in which Sb$_2$Se$_3$ was grown by exploiting the quasi-rheotaxia phenomenon, conversely shows a very high $TC_{310}$, almost negligible in other samples, except for sample C. As depicted in Figure 5b, the growth along the [001] direction is strongly window layer dependent, and it seems favoured by window layers containing zinc, reaching the maximum value of 27% for the texture coefficient of sample E.

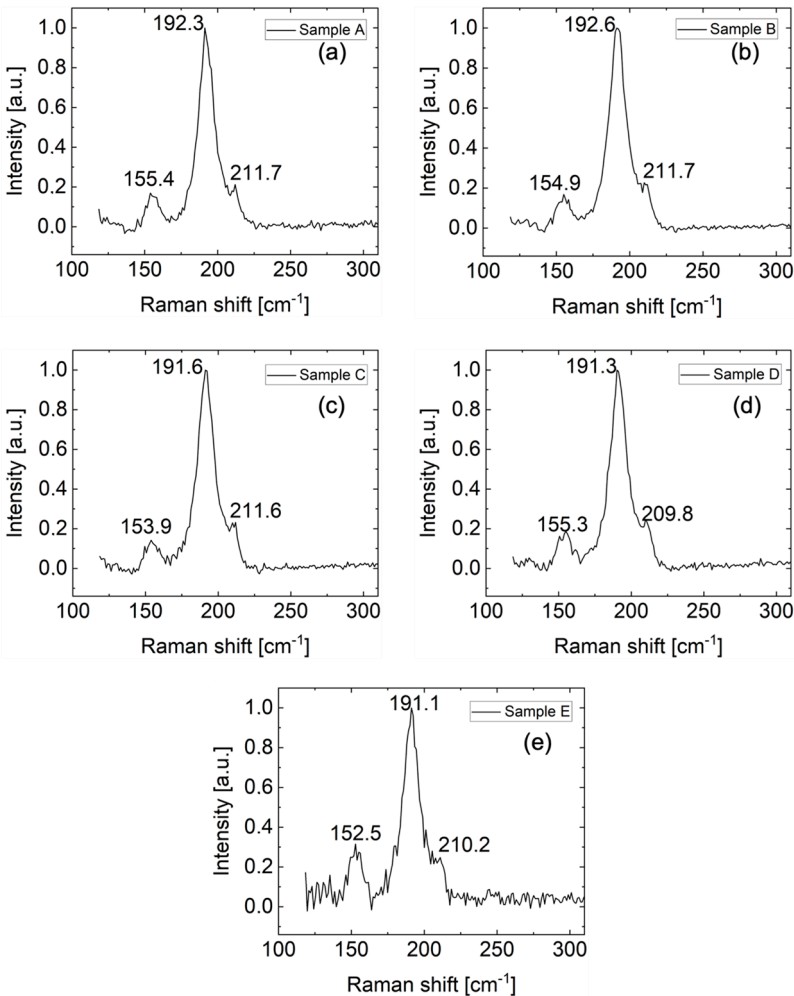

**Figure 4.** Antimony selenide Raman spectra grown on (**a**) CdS, (**b**) CdS:F, (**c**) CdSe, (**d**) As$_2$S$_3$ and (**e**) ZnCdS.

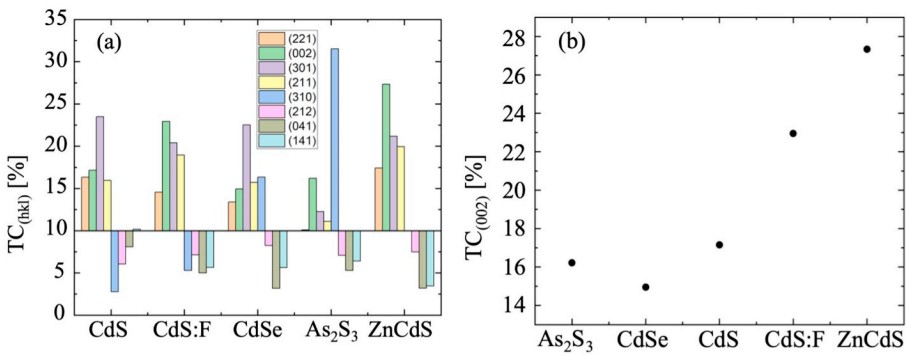

**Figure 5.** (**a**) Texture coefficient for different crystalline planes; histograms under the 10% line have to be considered as negative values with respect to 10%; (**b**) Trend of the texture coefficient of the (002) plane as a function of different window layers.

## 3.2. Electrical Characterization

To estimate the RT resistivity of the Sb$_2$Se$_3$ films, grown on an insulating ZnO layer, a 4-point measurement in the van der Pauw configuration was performed.

Referring to Figure 1c and permuting the pairs of contacts, it is possible to determine the resistivity from the following equation [44]:

$$\exp\left(-\frac{\pi d R_{12,34}}{\rho}\right) + \exp\left(-\frac{\pi d R_{23,41}}{\rho}\right) = 1$$

where:

- $d$ is the sample thickness
- $\rho$ is the resistivity
- $R_{12,34}$ is the average resistance over the possible pairs combinations of contacts $R_{12,34}$, $R_{21,34}$, $R_{34,21}$ and $R_{43,12}$. The same is true for $R_{23,41}$.

The resistivity resulted to be $5 \times 10^3 \ \Omega \cdot \text{cm}$ in dark conditions with an uncertainty of about 10%.

The J-V characteristics related to the complete devices with different window layers are reported in Figure 6a. From the J-V characteristics, it is possible to know the main photovoltaic parameters: the photocurrent $J_{sc}$ and the shunt resistance $R_{Sh}$, the photovoltage $V_{oc}$ and the series resistance $R_s$, are all estimated from the J-V curve.

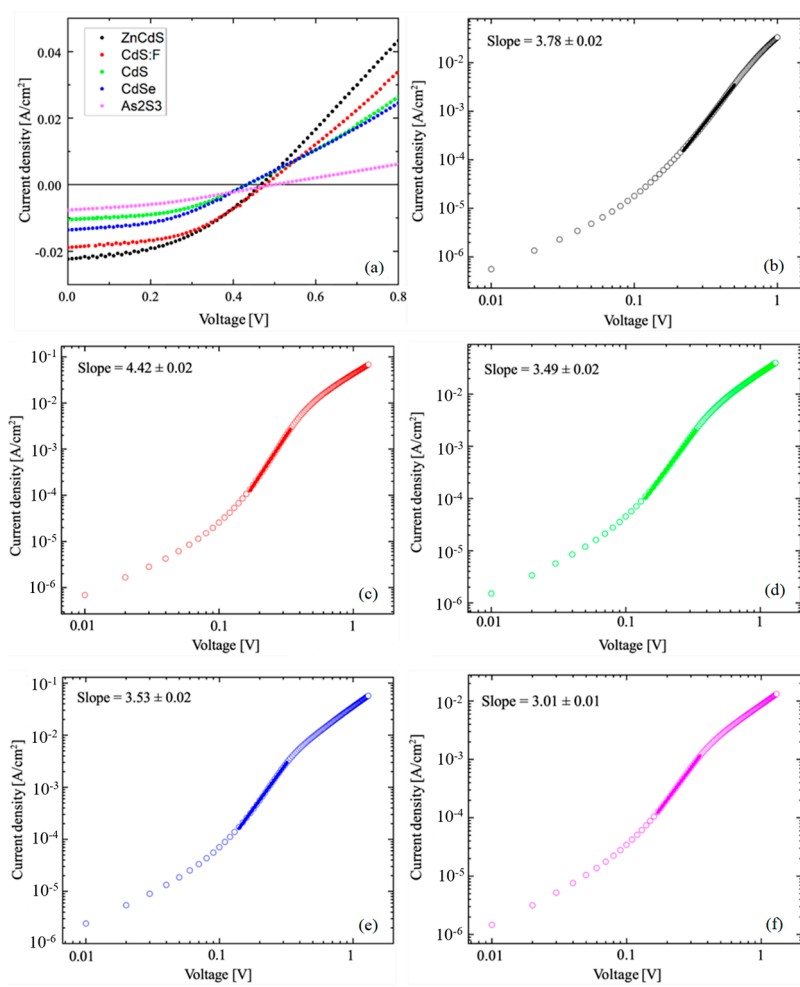

**Figure 6.** (**a**) J-V characteristics of the solar cells fabricated with different window layers; (**b–f**) Dark J-V curves in $\log_{10}$ scale. The fit (solid lines) was performed to extract the Trap Filling Limit Voltage (V$_{\text{TFL}}$) value (see text). (**b**) ZnCdS; (**c**) CdS:F; (**d**) CdS; (**e**) CdSe; (**f**) As$_2$S$_3$.

In order to determine the trap density of Sb$_2$Se$_3$ near to the top of the valence band and to the bottom of the conduction band [50], we have used the Space-Charge-Limited Current method. This method consists of measuring the J-V curve of the solar cell in dark

conditions with symmetric ohmic contacts [51,52]. The intermediate voltage region, where the slope of the curve is higher than 2, is the trap-filled limit region, suitable to determine the number of traps [53,54]. From Figure 6b–f, the value of the Trap Filling Limit Voltage ($V_{TFL}$) was evaluated by using the following equation [55].

$$N_{trap} = \frac{2\epsilon\epsilon_0 V_{TFL}}{qL^2}$$

where $\epsilon_o$ is the vacuum dielectric constant, $\epsilon$ is the $Sb_2Se_3$ relative dielectric constant (estimated to be 15.1) [54], $q$ is the electron charge and $L$ is the antimony selenide film thickness.

The measured photovoltaic parameters given in Table 2 are in agreement with the literature for this type of solar cell [56,57], except for sample D, which is characterized by a very high photovoltage and very poor photocurrent with high series resistance and a consequently low fill factor. This result has been considered a direct consequence of the crystallization of $Sb_2Se_3$ grown on arsenic sulphide well evidenced by the low texture coefficient.

**Table 2.** Main photovoltaic parameters of solar cells realized with different window layers. The experimental errors are lower than 2%.

| - | Sample A | Sample B | Sample C | Sample D | Sample E |
|---|---|---|---|---|---|
| $R_{Sh}\ [\Omega \cdot cm^2]$ | 139 | 107 | 98 | 124 | 86 |
| $R_s\ [\Omega \cdot cm^2]$ | 15 | 9 | 16 | 49 | 8 |
| $J_{sc}\ [mA/cm^2]$ | 10.5 | 18.9 | 13.6 | 5.2 | 22.4 |
| $V_{oc}\ [mV]$ | 435 | 478 | 433 | 593 | 469 |
| *Fill factor* | 0.45 | 0.46 | 0.43 | 0.34 | 0.43 |
| PCE | 2.1% | 4.2% | 2.6% | 1.3% | 4.5% |
| $N_{trap}\ [cm^{-3}]$ | $3.0 \times 10^{13}$ | $1.2 \times 10^{13}$ | $2.3 \times 10^{13}$ | $5.0 \times 10^{13}$ | $3.6 \times 10^{13}$ |

From the equation

$$\sigma = \rho^{-1} = ep\mu$$

it is possible to determine the hole density $p$ in dark conditions, using the obtained resistivity value of about $5 \cdot 10^3\ \Omega \cdot cm$ and, for the mobility $\mu$ a value of 5.1 cm$^2$ V$^{-1}$ s$^{-1}$ [3].

The obtained value for $p$ is about $2 \times 10^{14}$ cm$^{-3}$, about one order of magnitude higher than the estimated trap concentration. It is therefore necessary to reduce the number of traps and increase the hole spatial density to improve the solar cell behavior.

A connection between the texture coefficient TC and the photovoltaic parameters is inferred by plotting $J_{sc}$ versus the $TC_{(002)}$ corresponding to the [001] growth direction (Figure 7a). For samples B and E, where the preferential growth is along the [001] direction, the current increases with the texture coefficient referred to as plane (002), while for the other samples $J_{sc}$ is lower due to a more pronounced random arrangement of the grains. In fact, even if the $TC_{(002)}$ for the CdS window is higher than for CdSe, the photocurrent $J_{sc}$ is lower, because its flow depends also on the other planes, in particular (301) (preferential for both) and (221). The last consideration suggests that for obtaining a good $J_{sc}$, it is not sufficient that the $Sb_2Se_3$ film grows preferentially with the (002) plane parallel to the substrate ([001] is the growth direction) but also that the occurrence of other growth directions is limited in favor of the [001]. Fill factor and $V_{oc}$ do not exhibit such dependence because they are more strongly dependent on the high number of defects (traps) in the $Sb_2Se_3$ film.

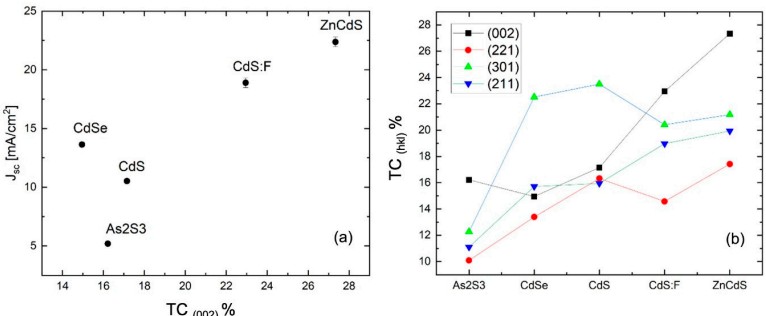

**Figure 7.** (**a**) Short circuit current density as a function of the $TC_{(002)}$ for different window layers; (**b**) TCs behavior of the main diffraction peaks for different window layers.

## 4. Conclusions

A systematic study of the interaction between different window layers (CdS, CdS:F, CdSe, $Zn_{0.15}Cd_{0.85}S$ and $As_2S_3$) deposited by sputtering and CSS-deposited $Sb_2Se_3$ was carried out. For the first time, the coupling between the materials of the window and absorber layers, deposited by the chosen techniques, was correlated.

$Sb_2Se_3$ films, grown on different window materials, show similar compositional and morphological properties but different preferential grain orientations, as evidenced by XRD and Raman measurements.

The combination of these techniques reveals that $Sb_2Se_3$ thin films do not present secondary crystalline phases. Through the texture coefficient evaluation, a trend between the $TC_{(002)}$ and the window layers on which the $Sb_2Se_3$ films were grown has been evidenced. In particular, the preferential $Sb_2Se_3$ growth direction is the [001] if *CdS : F* and *ZnCdS* are used as window layers.

From the J-V characteristics, the main photovoltaic parameters were extrapolated, and a trend of $J_{sc}$ vs. $TC_{(002)}$ was observed; in particular, $J_{sc}$ increases with the TC value for the (002) plane. For the other samples, a strong contribution from the other planes was observed. The other photovoltaic parameters, such as $V_{oc}$ and FF, are also strongly affected by the high number of traps in the antimony selenide film. Due to this fact, it is not possible to evaluate a substantial trend between the $TC_{(002)}$ and these parameters.

Although, for a further increase in the efficiencies on those devices, more studies are needed regarding window layers aiming to reduce $N_{trap}$ by adjusting the lattice mismatch at the heterojunction interface.

**Author Contributions:** Conceptualization, A.B.; Validation, D.S., A.P., S.M. and A.B.; Investigation, S.P., G.F., S.M. and S.V.; Data curation, S.P.; Writing–original draft, S.P. and G.F.; Writing – review & editing, D.S., S.M., R.F. and A.B.; Supervision, A.P., R.F. and A.B. All authors have read and agreed to the published version of the manuscript.

**Funding:** The activity carried out from RSE has been financed by the Research Fund for the Italian Electrical System under the Contract Agreement between RSE S.p.A. and the Ministry of Economic Development-General Directorate for the Electricity Market, Renewable Energy and Energy Efficiency, Nuclear Energy in compliance with the Decree of 16 April 2018.

**Institutional Review Board Statement:** Not applicable.

**Informed Consent Statement:** Not applicable.

**Data Availability Statement:** Not applicable.

**Acknowledgments:** The authors wish to thank Laura Fornasini and Danilo Bersani at the Department of Mathematical, Physical and Computer Science–University of Parma, for useful discussion on Raman measurements.

**Conflicts of Interest:** The authors declare no conflict of interest.

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
