# Peer review of "Sb2Se3 Polycrystalline Thin Films Grown on Different Window Layers"

_coatings, doi:10.3390/coatings13020338_

Round 1

Reviewer 1 Report

The authors showed the interaction between different window layers and antimony selenide. In particular, CdS, CdS:F, CdSe, As2S3 and ZnCdS thin films, deposited by radiofrequency magnetron sputtering have been tested as window layers. This manuscript is well written, but should be improved in several parts:

1.     The novelty of this manuscript is not clear enough. The authors should further highlight the novelty.

2.     The quality of some figures is not high. The authors should further improve the quality of figures for this manuscript (e.g., Figure 3).

3.     Can the proposed strategy extend to LEDs? The authors should explain it in detail, which will make this manuscript more interesting (Materials 2018, 11, 1376).

4.     How to further improve the performance by using the proposed strategy? The authors are suggested to give some comments.

5.     The format of references should be carefully checked. There are some mistakes.

6.     To make this manuscript more interesting and general, some papers should be cited (e.g., Cell Reports Physical Science 2022, 3, 100860).

Reviewer 2 Report

The paper is written well, especially the texture coefficients, but the reviewer needs some more highlights on the following things

1.      The research title is an essential element of the research paper. The current title is not well-written and must be brief and precise.

2.      The authors need to improve the abstract section as it does not look influencing and should also mention the performance of these window layers in one or two lines.

3.       Not only the Deposition method. Making powder Sb2Se3, the first step of the Sb2Se3, plays an essential role, for example, by which process it is made, vertical Bridgman method or other. Although the paper is not about the preparation of the solid-state method, it must be mentioned in the introduction

4.      In the materials and methods section, first give the information about samples A, B, C,…..

5.      In the XRD spectra (Fig. 3), all peaks are not defined, and for Raman spectra plots on the y-axis, the scale is not uniform for all figures.

6.      The authors should mention and cite these layer's performances for Sb2Se3-based solar cells with the help of previous studies [https://doi.org/10.3390/su132112320].

7.      For the better performance of a solar cell, the band gap and thickness also play a vital role. Authors should provide the band gap for these window layers with the help of UV-Vis characteristics.

8.      The distance of the substrate to the boat in case of close space sublimation is an important parameter and can be discussed in two or three lines.

9.      The author has mentioned, "Since this material crystallizes in the orthorhombic phase". The author needs to give more details and explanations about it.

10.  In Raman analysis, a 632.8 nm source is used, while another laser source is used in other literature. Kindly justify why 632.8 nm was used in the experiment.

 11.  Which one of the buffer layers provides the most stable device can also be mentioned as stability is one of the critical parameters for a solar cell.
The following papers can be cited
https://doi.org/10.1016/j.solener.2022.11.033

Reviewer 3 Report

High-performance crystalline films are obtained by optimizing thin-film growth conditions, resulting in high-performance solar cells. But this paper still has errors and unclear.

1. What is the size of the cell and what is the specific short-circuit current?

2. On page 5, line 142, the thickness of 4÷5um is entered incorrectly.

3. the TCO film is made of ITO glass after 800nm, in fact, the conductivity and transparency of ITO glass of 100-150nm is the best. Here 800nm ITO glass as transparent electrode, its light transmission is already very low.

4. All samples need to give the photoelectric conversion efficiency value.
